# Interplay of Multiple Sediment Routing Systems Revealed by Combined Sandstone Petrography and Heavy Mineral Analysis (HMA) in the South Pyrenean Foreland Basin

Xavier Coll [1] , Marta Roigé [1], David Gómez-Gras [1,*], Antonio Teixell [1], Salvador Boya [1] and Narcís Mestres [2]

1   Departament de Geologia, Universitat Autònoma de Barcelona, 08193 Bellaterra, Spain;
    xavier.coll.c@gmail.com (X.C.); roige.marta@gmail.com (M.R.); antonio.teixell@uab.cat (A.T.);
    salvaboya@gmail.com (S.B.)
2   Institut de Ciència de Materials de Barcelona, ICMAB-CSIC, Campus de la UAB, 08193 Bellaterra, Spain;
    narcis@icmab.es
*   Correspondence: david.gomez@uab.cat

**Abstract:** Combined sandstone petrography and heavy mineral analysis allow to decipher different sediment routing systems that could not be resolved by one method alone in the South Pyrenean foreland basin. We apply this approach to deltaic and alluvial deposits of the southern part of the Jaca basin, and in the time equivalent systems of the nearby Ainsa and Ebro basins, in order to unravel the evolution of source areas and the fluvial drainage from the Eocene to the Miocene. Our study allows the identification of four petrofacies and five heavy-mineral suites, which evidence the interplay of distinct routing systems, controlled by the emergence of tectonic structures. Two distinct axially-fed systems from the east coexisted in the fluvial Campodarbe Formation of the southern Jaca basin that were progressively replaced from east to west by transverse-fed systems sourced from northern source areas. In the late stages of evolution, the Ebro autochthonous basin and the Jaca piggy-back basin received detritus from source areas directly north of the basin from the Axial Zone and from the Basque Pyrenees. Coupling sandstone petrography with heavy mineral provenance analysis allows challenging the existing model of the South Pyrenean sediment dispersal, highlighting the relevance of this approach in source-to-sink studies.

**Keywords:** provenance; sandstone petrography; heavy minerals; sediment routing systems; Jaca basin; South Pyrenean foreland; Pyrenees

## 1. Introduction

The sedimentary record of a foreland basin offers the opportunity to study the interplay between distinct source areas, allowing to infer the uplift and exhumation history of mountain belts [1–3]. Sediment provenance analysis is a useful tool to understand the processes occurring in the hinterland of a sedimentary basin and enables to constrain the timing of geodynamic events, as well as to unravel sediment pathways and correlate stratigraphic sequences [4–10]. This arduous task requires combining as many provenance indicators as possible in order to achieve the highest resolution for identifying and characterizing the sediment routing systems in the related basins [8,11–15]. Sandstone petrography and heavy mineral analysis are widespread techniques in sedimentary provenance studies [5,8,16–22]. Since each of these methods can record different provenance signals, the integration of both is crucial to fully characterize and understand the functioning of sediment routing systems.

The deltaic to fluvial–alluvial sedimentary record of the South Pyrenean foreland basin (SPB) records stages of strong exhumation of the Pyrenean mountain belt. From mid Eocene to early Miocene times, these sediments were deposited in thrust-sheet-top basins featuring a wide range of lithologies from diverse sources [23–31]. In the western sector of the South Pyrenean foreland, the growing of the External Sierras thrust system from late

Eocene times (Figure 1) caused the compartmentalization of the Jaca thrust-sheet-top basin to the north and the autochthonous Ebro basin to the south [23,32–34].

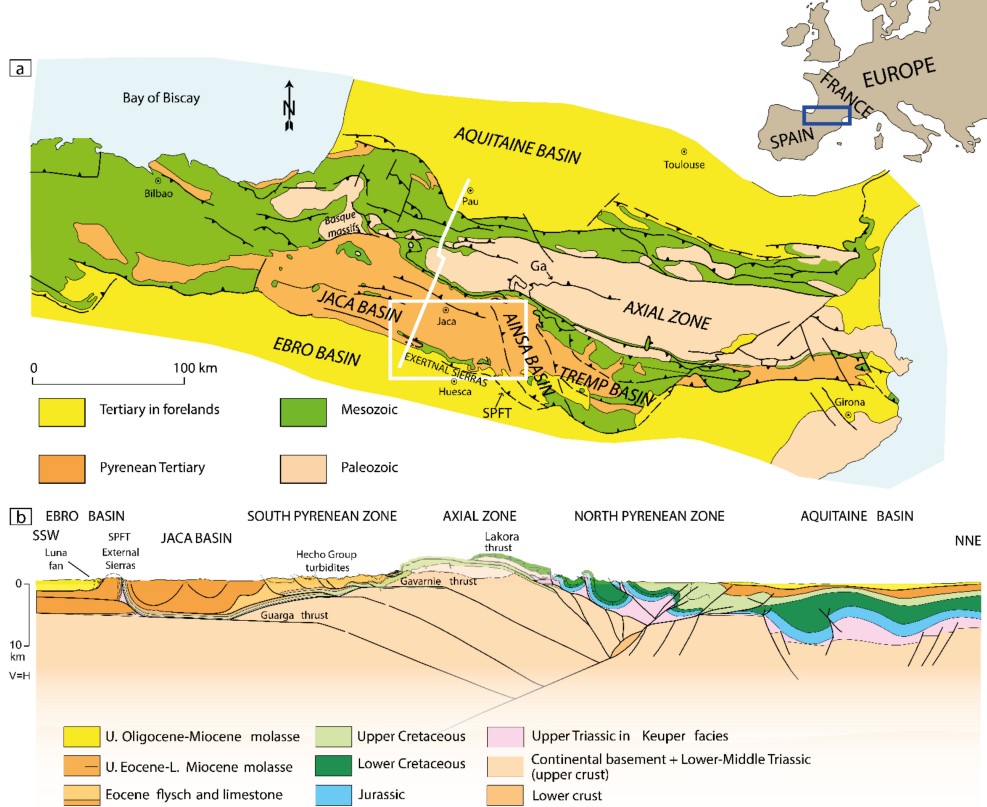

**Figure 1.** (**a**) Simplified geological map of the Pyrenees (redrawn from Teixell et al. [35]), showing the location of the study area (white frame). White line indicates cross-section in Figure 1b. White box corresponds to study area. Ga: Gavarnie thrust, SPTF: South Pyrenean Frontal Thrust. (**b**) Crustal cross-section of the west-central Pyrenees (simplified from Teixell et al. [36]), showing both the South Pyrenean Zone and the North Pyrenean Zone.

Previous provenance studies focused in the northern sector of the Jaca basin [29,30,37–39] or in the eastern Ainsa and Tremp-Graus basins (Figure 1) [28,31,40–43], Nevertheless, the sediment provenance (i.e., sandstone petrography, heavy minerals) of the southern part of the Jaca basin remain unknown.

The use of heavy mineral analysis as an effective provenance tool to unravel sediment sources or sediment pathways has been widely demonstrated in the Alps or the Himalayas, mainly on modern sediments [8,16,44–51]. However, few studies characterized the heavy mineral provenance signatures of the South Pyrenean and Ebro basins [28,39,40,52–58] and none have focused on the southern margin of the Jaca basin. In addition, the works integrating their results with a solid compositional framework based on sandstone petrography detrital modes are scarce. This contrasts with the amount of fruitful works that characterized the stratigraphy, sedimentology, magnetostratigraphy, paleontology, and tectonic structure of this area [23,24,59–79]. These provide a solid stratigraphic framework to characterize the sediment of source areas and the related routing systems, crucial to track the overarching evolution of the foreland basin in relation to the tectonic development of the Pyrenean orogenic belt.

In this work, we aim to unravel the compositional nature of the transitional to alluvial environments of the southern part of the Jaca basin through a provenance study that integrates sandstone petrography and heavy mineral analysis, constraining the interplay between the different active source areas that supplied the axial and the transverse systems. The time-equivalent deposits in the Ainsa basin (Escanilla Formation) are also analyzed

in order to test the connectivity between both basins. The Ebro basin deposits that occur south of the External Sierras thrust front are also investigated to compare their sediment provenance with those of the Jaca thrust-sheet-top basin, providing new insights into the last stages of the terrestrial sedimentation.

This work highlights the importance of coupling sandstone petrography and heavy mineral analysis in order to constrain sediment provenance and sediment dispersal patterns, with applications to collisional orogens where different source areas can produce similar compositional signatures. Our results demonstrate that a provenance framework based on a single technique can lead to biased conclusions and can overlook important details of the sediment routing into a clastic basin.

## 2. Geological Setting

### 2.1. Structural and Stratigraphic Framework

The Pyrenean fold-and-thrust belt grew diachronously, from late Cretaceous to Miocene, as a result of the oblique character of the collision between the Iberian and European plates [80–83]. The subduction of the lower crust of the Iberian plate under the European plate led to the inversion of the former Mesozoic rift basins and the stacking of the basement, resulting in an upper-crustal doubly-vergent orogenic prism. The core of the belt (known as the Axial Zone) is made of basement-involved stacked thrust sheets flanked to the north by a series of inverted hyper-extensional Mesozoic basins (the North Pyrenean Zone (Figure 1) [84]). The tertiary foreland deposits that occur further north constitute the Aquitanian basin. By contrast, the deformation in the southern Pyrenees was accommodated by a thrust imbricate fan [85–87], which in the west central Pyrenees comprise four main thrust sheets (Lakora-Eaux-Chaudes, Gavarnie, Broto, and Guarga). These thrust sheets involve the Paleozoic basement, a preorogenic Mesozoic succession, and the late Cretaceous to early Miocene foreland basin. In the Jaca area, the proximal basin was detached constituting a wedge-top basin, bordered to the south by the thrust front of the External Sierras (Figure 1b). South of the thrust front, the autochthonous Ebro basin exposes a young sedimentary record of the final stages of the Pyrenean exhumation.

Variscan low grade metamorphic rocks and granitoids comprise the major part of the Paleozoic basement (the core of the Pyrenean belt), which are, in turn, unconformably overlain by Permo–Triassic red beds or Cretaceous limestones. The preorogenic Mesozoic succession starts with the Triassic Keuper facies, which are involved in thrust sheet propagation, acting as an evaporite detachment level during extension and contraction, and salt diapirism processes that played a critical role in the formation of Mesozoic minibasins and the exhumation of the Paleozoic basement in the central Pyrenees [88,89]. The rest of the succession is made up of a thick Jurassic–Cretaceous carbonate and sandstone-shale successions.

The South Pyrenean basin contains synorogenic deposits of late Santonian to early Miocene age. During the Eocene, fluvio-deltaic sedimentary environments were concentrated in the Àger and Tremp-Graus basins (eastern sector of the South Pyrenean basin), funneling sediments to the west, to the slope, and deep-marine sedimentation environments of the Ainsa and Jaca basins [24,27,29,90,91]. This deep-marine succession (known as the Hecho Group turbidites [91]) developed during an underfilled foreland basin stage, which with the growth of the orogen, was progressively replaced, from east to west, by deltaic and alluvial deposits leading to an overfilled foreland basin stage (mid to late Eocene [23,25]).

The deep-marine deposits were replaced by the deltaic and fluvial Sobrarbe (Lutetian–Bartonian) and Escanilla (Bartonian–Priabonian) Formations in the Ainsa basin [92]. These systems, which have their source area located in the central Pyrenees and received sediments from the Sis and Gurp-Pobla paleovalleys, prograde westward into the Jaca basin [24,28]. The youngest deposits preserved in the Ainsa basin are the Graus Formation conglomerates (Chattian–Aquitanian), unconformably overlying the Escanilla Formation [93]. In the Jaca basin, the time equivalent deposits to the Escanilla and Sobrarbe are the Sabiñanigo Sandstone (Bartonian) and Belsué-Atarés (Bartonian–Priabonian) delta formations, the fluvial

Campodarbe formation (Bartonian–Oligocene), and the Bernués formation (Oligocene–Miocene), which record the evolution from transitional to fully terrestrial environments. This transition is diachronic and ends with the onset of the endorheic basin stage at 36 Ma, when terrestrial environments spread throughout the entire basin [66,71,94,95].

The Belsué-Atarés delta prograded from east to west at the same time that obliquely-trending folds started to grow in the southern margin of the Jaca basin [23,75,96]. These produced marked thickness variations and a strong diachrony of the sedimentary record. In the north-western part of the Jaca basin, a subordinated delta system occurs instead of the Belsué-Atarés delta (the Martés sandstone [23]). These deltaic environments were progressively substituted by the fluvial to alluvial Campodarbe and Bernués Formations spanning until the lower Miocene [23,76,97].

The Campodarbe Formation [60] is a fluvial to alluvial succession, where at least two main sediment routings can be identified [23]. In the northern margin, an east-derived axial fluvial system, entering the basin trough the south-eastern margin, interacts with a north-derived transverse alluvial fan system, mainly controlled by the activity of the Gavarnie thrust, and mostly derived from the recycling of a former lower to middle Eocene turbidite basin [23,29,30,97]. By contrast, the sedimentation in the southern edge was dominated by axially-fed fluvial systems that have its proximal time equivalents in the nearby Ainsa basin (Figure 1) (Escanilla Formation) and was strongly controlled by the growth of tectonic structures [23,24,32,63,65,67,75,98]. The last stages of the basin infill are marked by the Bernués Formation (Chattian–Aquitanian [23,97,99]), a complex of alluvial fan deposits sourced from the existing reliefs to the north of the basin.

As the orogenic deformation progressed to the south, the External Sierras thrust front [35,59,74,85] became strongly emergent (Oligocene–Miocene) and split the Campodarbe Formation in the Jaca basin to the north from the Ebro basin to the south. In the Ebro basin, the top of the Campodarbe Formation has been dated at 24.5 Ma (Chattian) [74,77], whereas the fluvial/alluvial deposits of the overlying Uncastillo Formation have been dated as Chattian–Aquitanian. The activity of the Guarga thrust sheet triggered the formation of this north-derived Luna alluvial fan system, which is sourced from the recycling of the Jaca basin and the existing reliefs of the Axial Zone farther to the north [23,100,101].

*2.2. Source Rock Lithologies*

The potential source areas for the Jaca basin during the late Eocene–Miocene are: (i) the Paleozoic basement of the Axial and North Pyrenean Zones, (ii) the preorogenic Mesozoic cover succession, and (iii) the earlier synorogenic assemblage of the upper Cretaceous to middle Eocene deposits (Figure 2).

The Paleozoic basement is constituted by Variscan granitoids that intrude an assemblage of Cambro–Ordovician (and locally Neoproterozoic) to Devonian metasedimentary units, which are, in turn, overlain by flysch deposits of Carboniferous (Culm facies). In the Eastern Pyrenees, the Cambro–Ordovician metasedimentary units dominate the present-day outcrops, intruded by the Ordovician orthogneisses and Variscan granitoids [102–106]. In this area, scarce Neoproterozoic outcrops are also present, mainly constituted by schists, limestones, dolomites, and migmatites. By contrast, in the central Pyrenees, the metasedimentary Devonian terrains (mainly limestones) coexist with the Cambro–Ordovician metasiliciclastic rocks. This metasedimentary succession is also intruded by Variscan granitoids, but Ordovician Orthogneiss do not occur in this area. In the western Axial Zone, Devonian and Carboniferous rocks are dominant with Cambro–Ordovician terrains almost non-existent. In general, the entire Paleozoic basement displays a very-low to low grade metamorphism, though it can reach the medium and high grade in the metamorphic domes that occur along the Axial Zone, mainly involving the Precambrian and Cambro–Ordovician terrains.

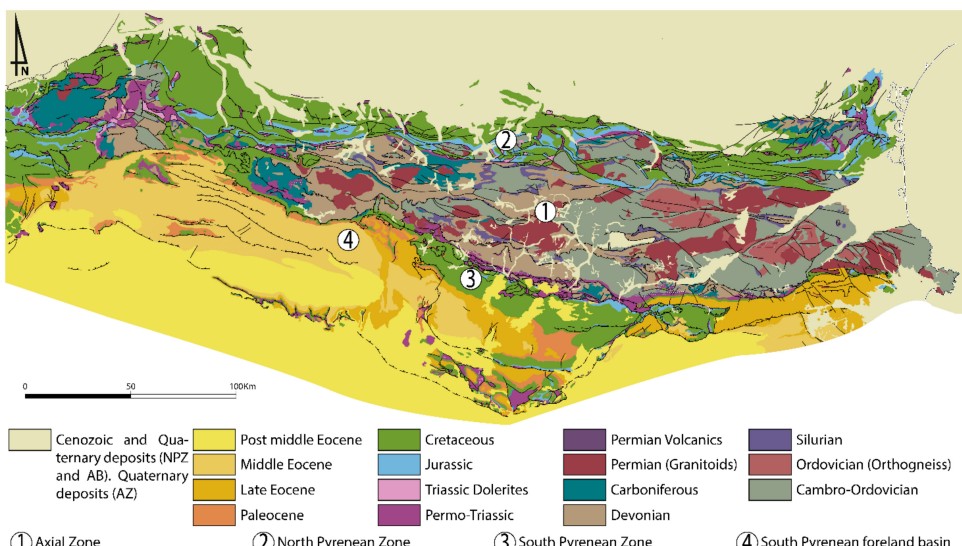

**Figure 2.** Geological map of the central Pyrenees (modified from a synthesis by Rodríguez-Fernández et al. [107]) showing the potential source rock terrains for the late Eocene–Oligocene systems of the Jaca basin. Dark frame represents the location of the study area. AB: Aquitanian basin; NPZ: North Pyrenean Zone; AZ: Axial Zone.

The Paleozoic basement of the Axial Zone is unconformably overlain by Permo–Triassic red beds or Jurassic–Cretaceous carbonates. Shales, carbonates (Muschelkalk facies), evaporites (Keuper facies), and dolerites (ophites) follow the lower Triassic sandstone. In the North Pyrenean Zone, Jurassic-lower Cretaceous carbonate deposits are followed by a thick shale and turbidite sequence (Albian to Maastrichtian) intruded by subvolcanic basaltic rocks [108,109]. Contrary to what occurs in the South Pyrenean Zone, Paleozoic basement (locally Neoproterozoic) outcrops also occur, mainly concentrated in the east-central and western Pyrenees. In addition, a restricted narrow east-west-trending belt (the Internal Metamorphic Zone) displays a HT-LP metamorphism related to crustal thinning and mantle exhumation during the Cretaceous rifting [110,111], mainly affecting the Jurassic and Cretaceous succession. In contrast, in the South Pyrenean Zone, the Jurassic, the Cretaceous, and part of the early foreland-basins deposits consist of platform limestones, dolostones, and sandstones. These deposits developed in the distal margin of the marine foreland basin from late Cretaceous to Lutetian times, whereas the basin trough was characterized by clastic deposits.

*2.3. Heavy Minerals and Source Rock Lithologies*

The heavy-mineral content of a sedimentary rock usually does not exceed 1% of the total volume. Source rock type and fertility are the primary controls on the heavy mineral content that a source can provide [19,112,113]. Igneous and medium to high grade metamorphic rocks can contain various heavy minerals as their main constituents or accessory phases, whereas siliciclastic sedimentary rocks mainly produce recycled ultrastable minerals. By contrast, marine carbonate rocks are usually devoid of heavy minerals, although they may produce a few recycled minerals, originally incorporated by aeolian input or by diluted suspended material from terrestrial sources.

In the Pyrenees, zircon, tourmaline, rutile, and apatite grains occur in a wide variety of igneous, sedimentary, and metamorphic rocks of the Paleozoic basement, Mesozoic metamorphic rocks, and Mesozoic and Tertiary sedimentary cover (Figure 2). Paleozoic metapelites such as phyllites, schists, and granulites might contain chloritoid, almandine, staurolite, and kyanite [102,103,105]. However, these minerals have never been reported in the Mesozoic metapelites of the North Pyrenean Zone. Permo–Carboniferous igneous rocks described in the Pyrenees (Carboniferous rhyolites, dacites, ignimbrites, volcaniclastic

sediments [114,115] or late Variscan muscovite granites [116]) can also be a source of almandine garnet. By contrast, grossular garnet is usually associated with skarn deposits, thermally metamorphosed impure limestones, and marbles occurring in the Axial and North Pyrenean Zones, although volcanic rocks of the North Pyrenean Zone (syenites) may contain grossular as well [117]. Clinopyroxene, olivine, spinel, and epidote have been described in various igneous rocks such as Triassic dolerites or Cretaceous basalts, picrites, teschenites, syenites, and lamprophyres of the North Pyrenean Zone [117–120]. Clinopyroxene also occurs commonly in basaltic and andesitic rocks of the Stephano-Permian vulcanism [114]. In addition, epidote, titanite, and clinopyroxene can be found in Paleozoic marbles and calcschists, skarn deposits, and hornfels related to Paleozoic granites, as well as in the metamorphic Mesozoic limestones of the North Pyrenean Zone [119,121]. In regionally metamorphosed carbonate rocks of the amphibolite facies, spinel, olivine, clinopyroxene, and amphibole have been reported. Titanite, a common accessory mineral of many igneous and metamorphic rocks, can be found in Paleozoic granitic sources, Triassic dolerites, metapelites, and impure calc-silicate rocks [102,103,105,106,109,118].

## 3. Sampling and Analytical Methods

The deltaic to fluvial/alluvial environments of the Belsué-Atarés, Campodarbe, and Bernués Formations were sampled along 6 stratigraphic sections of the southern part of the Jaca basin (San Felices, Rodellar-Bibán, Monrepós, Gállego, Salinas, and Martés sections) (Figure 3). In addition, a section in the Ebro basin (Luesia section, covering the Campodarbe and Uncastillo Formations) and another one in the Ainsa basin (Ainsa section, including the Sobrarbe and Escanilla Formations) were also sampled to compare the compositional features with the time-equivalent deposits of the Jaca basin.

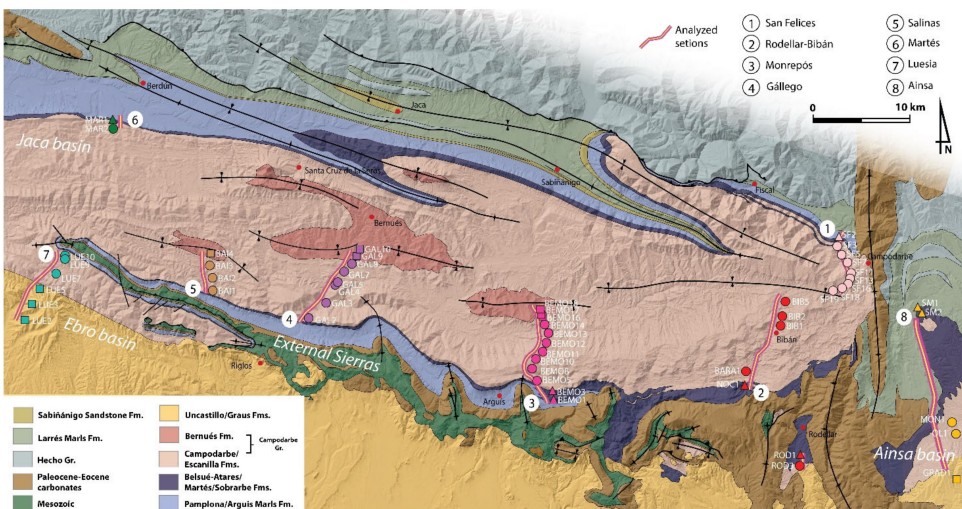

**Figure 3.** Geological map of the of the Jaca basin (modified from Puigdefàbregas [23]). Yellow-purple lines show the location of the analyzed sections. Numbers refer to each section: (1) San Felices section, (2) Rodellar-Bibán section, (3) Monrepós section, (4) Gállego section, (5) Salinas section, (6) Martés section, (7) Luesia section, and (8) Ainsa section. Numbers and symbols refer to each of the analyzed samples for sandstone petrography analyses.

Ninety-four sandstone and conglomerate samples were collected in the field for petrography and heavy mineral analysis. The number and spacing of petrography samples were established according to the representativeness of each analyzed sedimentary system within the stratigraphic sections. Fifty-three samples were chosen for quantification of the detrital modes through point-counting analysis under the polarizing microscope. After establishing a provenance framework based on sandstone petrography, twenty-five samples were selected for heavy mineral analysis.

### 3.1. Sandstone Petrography

The petrographic study was carried on thin sections stained with Na-cobaltinitrite for suitable identification of feldspar [122]. In order to distinguish carbonate compositions such as dolomite, ankerite or calcite, Alizarine red-S staining was applied. The Gazzi–Dickinson point counting method [1,123–125] was used to calculate the detrital modes, counting three to five hundred points for each thin section [126]. The points were classified as framework grains, diagenetic minerals, matrix, and porosity. Framework grains were labeled according to Zuffa [125] as noncarbonate extrabasinal (NCE), noncarbonate intrabasinal (NCI), carbonate extrabasinal (CE), and carbonate intrabasinal (CI). For metamorphic rock fragments, the classification of Garzanti and Vezzoli [127] was applied, while volcanic grains were classified according to Margsaglia and Ingersoll [128] and Critelli and Ingersoll [129]. The results were plotted and classified into first to fourth order ternary diagrams following Zuffa [125].

### 3.2. Heavy Minerals

Medium grained sandstone samples were targeted for sampling in order to avoid hydraulic-sorting effects that might bias the analytical results [130–133]. Fine to very coarse grain sizes were collected only in cases where medium grained sandstone was not available. In addition, samples from each depositional system were collected from similar facies in order to minimize hydraulic-sorting effects related to different processes within the same depositional environment.

Samples were crushed with a Retsch Disc Mill DM 200 prior to acid digestion with diluted 10% acetic acid for carbonate removal and avoiding apatite dissolution [112]. Struers Metason 200 ultrasound machine was used during 5 min in order help desegregation of well cemented sands and clay coatings. The 32 to 500 micrometer window was obtained through wet sieving, in order to avoid the clay to fine silt fraction but to analyze an acceptable grain-size window that does not produce a potential bias due to hydraulic-sorting effects [130–132]. The recovery of the dense fraction (2.90 g/cm$^3$) was performed by the centrifuging method, using the nontoxic dense liquid Na-polytungstate and partial freezing with liquid nitrogen [112,133]. 30 µm polished thin sections of the heavy-mineral fraction were prepared for each sample.

We used Raman spectroscopy for the identification of mineral grains [134,135]. A representative area of each thin section was selected and at least 200 non-diagenetic transparent heavy minerals were analyzed (opaque, carbonate, and micaceous minerals were not considered for identification) [112,136,137]. Therefore, only relative abundances of heavy minerals are reported in this paper. Raman scattering experiments were performed at room temperature in the backscattering geometry using a T64000 Horiba Jobin Yvon micro-Raman spectrometer equipped with high sensitivity liquid Nitrogen cooled CCD (charge-coupled device) as the detector. Samples were mounted on the XY stage of a BX40 Olympus microscope. The 488 nm laser line was used for the measurements. The incident laser beam was focused to a 2 µm spot on the samples using a 50-microscope objective. Laser power was kept below 0.4 mW to avoid laser-induced heating. Spectrometer resolution was 2 cm$^{-1}$. The obtained spectra (Figures S1 and S2) were compared with reference spectra [135,138,139] and mineral identification was verified under the optical microscope.

### 3.3. Statistical Treatment

In this work, we apply correspondence analysis [140] as an exploratory compositional data analysis tool to assess similarities between samples. The results are displayed as biplots in order to facilitate the visualization and interpretation of the results.

Statistical treatment of the point-counting data (petrographic and heavy mineral data) was performed using the Provenance R-package [141,142], which allowed the distinction between different petrofacies and heavy-mineral suites. Since the heavy-mineral data were acquired using the area method, the statistical bias might be greater; however, we believe it is not significant for the purpose of this work.

## 4. Results

### *4.1. Sandstone Petrography*

#### 4.1.1. Grain Types

Framework grains are here described in order to establish their most probable provenance. Non-framework grains are authigenic minerals, related to cementation and replacing processes in most of the cases where calcite is the main cement typology. All percentages here described are referred over total framework grains.

Noncarbonate Extrabasinal Grains (NCE)

Quartz is a widely represented type of grain. Its contents range from 7.3 to 51.9%. Several types of quartz have been distinguished: monocrystalline, polycrystalline, and quartz contained in a rock fragment. Characteristic quartz with evaporitic inclusions (anhydrite and halite) occurs in proportions of 0.4–2.6%.

Feldspar grain contents are (Figure S3) up to 12.4% of abundance, classified as orthoclase (<6.5%), microcline (<4.5%), and plagioclase (<5.5%). K-feldspar usually appears non-altered, whereas plagioclase usually shows some degree of alteration.

Lithic grains (Figure S3) dominate the framework components in many samples. They consist of metamorphic, plutonic, volcanic, and noncarbonate sedimentary rock fragments. Metamorphic rock fragments are the most abundant type of grain in most of the samples (up to 46.1%). Metamorphic grains include very low to low grade (metapelites and phyllites), medium grade (mica schists, schists, and chloritic schists), and high grade (quartzite). Plutonic grains (granitoid rock fragments) are very scarce (<1.7%) and recognized in very few samples. Volcanic grains represent up to 3.6% in some samples. Three textures of paleovolcanic lithics have been identified: (i) lathwork texture made of plagioclase and altered augite crystals, (ii) microlithic texture made of plagioclase microlites, and in lower proportion, (iii) vitric texture. Noncarbonate sedimentary rock fragments (3.4–53.6%) are sandstone, hybrid sandstone (Figure S3), siltstone, hybrid siltstone, and silicified rock fragments. Silicified rock fragments have been also subdivided into radiolarite rock fragments and silicified limestones.

Noncarbonate Intrabasinal Grains (NCI)

Noncarbonate intrabasinal grains are scarce (<5.3%), appearing always as glauconite or argillaceous rip-up clasts.

Carbonate Extrabasinal Grains (CE)

Carbonate extrabasinal grains occur with a wide variety of textures, reaching proportions up to 64.6%. Distinction has been made into (i) bioclastic and sparitic limestones, (ii) dolostones, and (iii) dolomitic and dolomitized limestones. Most common components contained in these rock fragments are bioclasts as foraminifera (nummulitids, discocyclinids, miliolids, alveolinids), red algae, or bivalves. Dolostone fragments (<6.7%) have been recognized as dolomicrite, polycrystalline sparitic fragments and single-grain dolomite.

Carbonate Intrabasinal Grains (CI)

Carbonate intrabasinal grains are rare (<4%) and appear as micritic intraclasts and caliche concretions, or as bioclasts (red algae, bivalves, and benthic foraminifera such as Nummulites).

#### 4.1.2. Modal Sandstone Composition

Sandstone detrital modes are classified in three ternary diagrams (Figure S4), in order to visualize the compositional trends and the potential shifts of the source areas. A first-order diagram is used to classify the analyzed samples according to Zuffa 1980 [143]. The results show that the analyzed samples correspond to lithic arenites and calclithites. A second-order classification diagram following Dickinson et al. [144] shows an increase in

lithic fragments from the deltaic to the alluvial environments. Finally, a third-order diagram shows the dominance of metamorphic and sedimentary over volcanic lithic grains.

### 4.1.3. Petrofacies

We use a ternary plot in order to discriminate among the petrofacies defined by Roigé et al. [30] for the northern Jaca basin. The plot (Figure 4a) compares the relative abundance of hybrid sandstone rock fragments (Hy.Sst), feldspar and lithic rock fragments, excluding hybrid sandstone rock fragments (F+L) and carbonate extrabasinal grains (CE). The discrimination between the relative abundance of these types of grains allows to define four petrofacies that reflect the interplay of different source areas and the evolution of the basin.

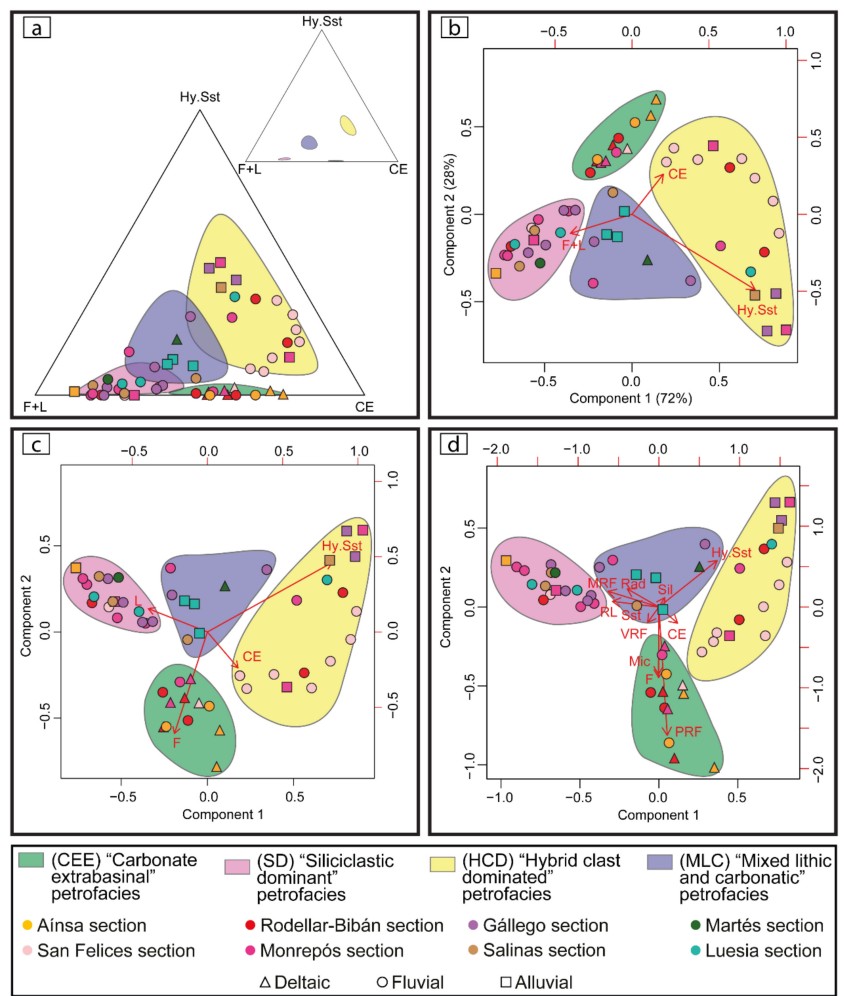

**Figure 4.** Compositional plots for all the analyzed samples. Feldspar (F), metamorphic r.f. (MRF), plutonic r.f. (PRF), volcanic r.f. (VRF), hybrid sandstone r.f. (Hy.Sst), siliciclastic sandstone (Sst), micas (Mic), silicified r.f. (Sil), radiolarite r.f. (Rad), Fe-Oxide replacement r.f. (RL), and carbonate extrabasinal grains (CE). (**a**) Compositional plot discriminates the four main groups of petrofacies described for all the analyzed samples showing the confidence region (90%) of the entire population of each petrofacies, while the small ternary diagram on the right side shows the mean confidence regions (90%) for each petrofacies. (**b**) Biplot showing the statistical significance of the four petrofacies model (100% of the variance is explained). (**c**) Biplot displaying the results of a correspondence analysis where F and L have been considered as two different variables in order to illustrate the compositional variations of feldspar (F) and lithics (L). (**d**) Biplot showing the compositions of samples and petrofacies considering a wide range of grains.

Correspondence analysis is here used in order to assess the statistical significance of the defined petrofacies. The results (Figure 4b) indicate that the four petrofacies model accounts for a hundred percent of the variance. Moreover, additional biplots differentiating F from L (Figure 4c) or showing the relative abundance of more types of grains (Figure 4d) are used for further description and visualization of samples' composition.

Carbonate Extrabasinal Enriched Petrofacies (CEE)

In this petrofacies (Figures 5A and S5a), carbonate extrabasinal grains are the most dominant rock fragment (48.5–75.3%). Lime mudstone and wackestone rock fragments of Mesozoic age are the most represented, including wackestone rock fragments containing *phitonellid* tests (Turonian limestones from the southern Pyrenees), while grainstone, packstone, and dolostone rock fragments are also present. This petrofacies also displays significant enrichment in microcline, orthoclase, plagioclase, plutonic rock fragments, and mica (Figure 4c,d), but it lacks hybrid sandstone rock fragments (<2.6%).

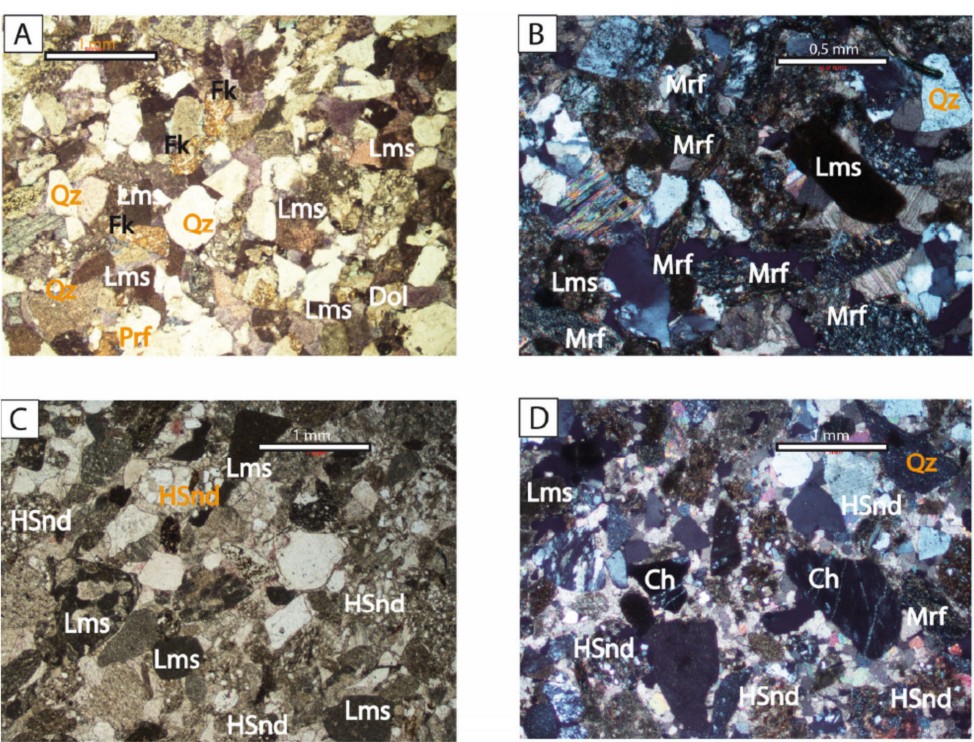

**Figure 5.** Optical photomicrographs of the described petrofacies. (**A**) General view of "carbonate extrabasinal enriched" petrofacies, with abundant micritic and bioclastic limestone (Lms) fragments and quartz (Qz), K-feldpar (Fk), plutonic rock fragments (Prf), and dolomite grains (Dol) (XPL). Sample ROD1, Belsué-Atarés Formation. (**B**) "Siliciclastic dominant" petrofacies characterized by the highest contents of quartz (Q), metamorphic rock fragments (Mrf), and limestone grains (Lms) and by the absence of hybrid sandstone rock fragments (XPL). Sample GAL4, Campodarbe Formation. (**C**) General view of "hybrid clast-dominated" petrofacies showing the large amount of hybrid sandstone rock fragments (HSnd) and limestone rock fragments (Lms) (PPL). (**D**) Appearance of "mixed lithic and carbonatic" petrofacies, showing the coexistence of hybrid sandstone rock fragments (HSnd) with abundant carbonatic (Lms) and siliciclastic grains radiolarite (Ch), quartz (Qz), and metamorphic grains (Mrf) (XPL).

In the interstratified conglomerate layers, the most common clast types are Mesozoic grey micritic limestones and dolostones. Epidote-bearing dolerites (Triassic ophites) are also present. Subordinate clasts are siliciclastic red sandstone (Permotriassic) and white quartz pebbles, green quartzite (Paleozoic), and black quartz clasts (Carboniferous radiolarites).

The CEE petrofacies occurs in the oldest analyzed sedimentary systems of the basin (the Sobrarbe and Escanilla Formations in the Ainsa basin, the Belsué-Atarés Formation and the basal Campodarbe Formation in the Jaca basin; Figure 6).

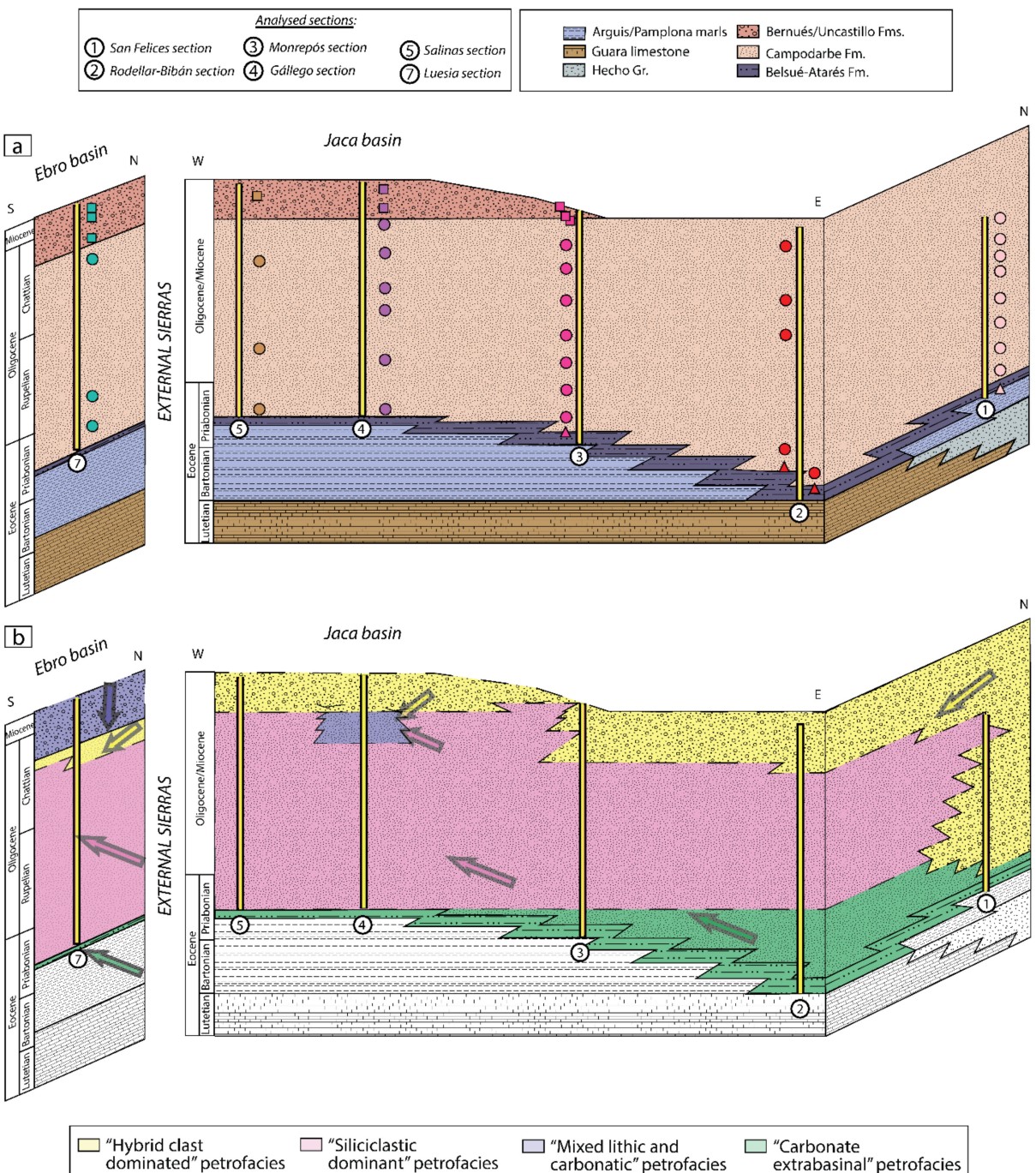

**Figure 6.** (**a**) General stratigraphic cross-section sketch with symbols representing the relative position of the analyzed samples presented in Figure 3, used here to understand the petrofacies scheme below. (**b**) Colored stratigraphic cross-section sketch in order to illustrate the distribution of the petrofacies laterally and through time. Yellow color corresponds to "hybrid clast-dominated" petrofacies, green color to "carbonate extrabasinal enriched" petrofacies, blue color to "mixed lithic and carbonatic" petrofacies, and pink color to "siliciclastic dominant" petrofacies. Colored arrows are used to facilitate reading of the provenance information. Dashed lines represent the boundaries between petrofacies.

The Sobrarbe Formation (samples SM1 and SM3 in Figure 3) displays the highest relative content of carbonate grains (mudstone, packstone, and grainstone rock fragments; Figure 4a), together with subordinate granite, schist, and feldspar grains. Up section, the overlying Escanilla Formation (samples MO1 and OL1) displays the same compositional features, with a slight increase in lithic fragments (Figure 4c).

In the Jaca basin, this petrofacies is observed in the Belsué-Atarés Formation in the San Felices, Rodellar-Bibán, and Monrepós sections, displaying a composition very close to that described in the Escanilla Formation in the Ainsa basin (Figure 4). In the Rodellar-Bibán section (samples ROD1 and ROD3), the petrofacies contains the highest abundance of feldspar grains (reaching up to 18.5% of framework grains).

Siliciclastic Dominant Petrofacies (SD)

Siliciclastic components are the most dominant grain type (Figures 5B and S5b,c), and the abundance of carbonate grains is always under 37%. Hybrid sandstone rock fragments are present in this petrofacies but are scarce (>5.5%). An enrichment of MRF is always observed (Figure 4d), together with significant amounts of radiolarite rock fragments and quartz-rich sandstone/siltstone rock fragments. Subordinate sandstone/siltstone rock fragments usually contain mica and opaques (RL).

In the conglomerate beds, the dominant clast types are siliciclastic red sandstone and microconglomerate pebbles (deriving from the Buntsandstein facies), together with low grade metamorphic clasts (slates and schists; Paleozoic), epidote-bearing dolerites (Triassic ophites), granitoids (Variscan), white quartz, green quartzites (Paleozoic), and black quartz clasts (Carboniferous radiolarites). Grey micritic limestones and dolostones (Mesozoic) and Devonian limestones are also present. Hybrid sandstone clasts are scarce.

In the Ainsa basin, siliciclastic dominant petrofacies are represented in the upper part of the Escanilla Formation (sample GRAD1; Figure 6), where metamorphic rock fragments such as metasiltstone, slate, and phyllite are overwhelming (Figure 4d).

In the Jaca basin, the Campodarbe and Bernués Formations display similar content to that of the upper part of the Escanilla Formation in the Ainsa basin. In the Rodellar and San Felices sections, this SD petrofacies is scarce. By contrast, to the west, the SD petrofacies dominates most of the Campodarbe Formation (Bibán, Monrepós, Gállego, Martés, Salinas, and Luesia sections; Figure 6). Only one sample of the Bernués Formation in the Monrepós section displays this petrofacies (BEMO-18).

Hybrid Clast-Dominated Petrofacies (HCD)

Limestone rock fragments are the most represented component in most of the samples of this petrofacies (>30.6%; Figure 4). However, the distinctive feature of this petrofacies (Figures 5C and S5d) is the high content (>8.4%) of hybrid sandstone/siltstone rock fragments (rock fragments that contain both extrabasinal and intrabasinal carbonate components in similar proportions) when compared to the other petrofacies. Silicic components such as metamorphic and siliciclastic sandstone rock fragments display a low abundance (<9.7 and <12.3%, respectively; Figure 4c,d).

In the conglomerate layers of this petrofacies, hybrid sandstone rock fragments (mainly derived from the recycling Eocene Hecho turbidites) and grey micritic limestones and dolostones (Mesozoic and lower Tertiary) dominate over the lithologies described in the "siliciclastic dominant" petrofacies (Permotriassic pebbles, low grade metamorphics, granitoids, white quartz, green quartzite, and radiolarites). Triassic dolerites have also been observed in the eastern sector of the basin. In the Bernués Formation (Gállego and Salinas section), clasts of metamorphic breccia (Ibarrondoa breccia; upper Cretaceous, North Pyrenean Zone) have been identified.

The HCD petrofacies dominates the Campodarbe deposits in the San Felices section (Figure 6), whereas in the south-west part of the basin, only the upper parts of this Formation (Rodellar-Bibán and Monrepós sections) record this petrofacies. The overlying Bernués Formation maintain these compositional features (Figures 4 and 6).

Mixed Lithic and Carbonatic Petrofacies (MLC)

This petrofacies (Figure 4a) is characterized by 20.9–45.9% carbonate grains (Figures 5D and S5e,f), including wackestone rock fragments containing *phitonellid* tests (Turonian limestones from the southern Pyrenees). It also contains, as noncarbonate grains, up to 27.5% hybrid sandstone rock fragments and high proportions of lithic grains (20.9–45.9%; excluding hybrid sandstone rock fragments) such as metamorphic r.f., radiolarite r.f., siliciclastic sandstone, and volcanic lithic grains, as well as K-feldspar.

This petrofacies is observed in a limited number of samples (Figures 4 and 6) and shows two different sub-groups that can be identified in function of the lithic fragment types (Figure S6). In the Campodarbe Formation, it appears toward the top in the Monrepós section and extends to lower stratigraphic levels in the Gállego and Salinas sections (samples BEMO14, GAL7, GAL8, BAI3). It shows a significant enrichment in metamorphic rock fragments (MRF) and Fe-Oxide replacement r.f. (RL) (Figure S6). In contrast, in the transitional Martés deposits (sample MAR1) and in the alluvial Uncastillo Formation (samples LUE2, LUE3 and LUE5), lithic grains are enriched in silicified r.f. (Sil), radiolarite r.f (Rad), and volcanic r.f. (VRF).

The conglomerate layers of the Campodarbe Formation in the Gállego and Salinas sections display hybrid sandstone clasts (mainly Hecho Group turbidites; Eocene), low grade metamorphic clasts (slates and schists; Paleozoic), Permotriassic red sandstones, Devonian limestones, and grey micritic limestones and dolostones (Mesozoic). In the Monrepós area, epidote-bearing dolerites (Triassic ophites) are also present. In the Luesia section, the conglomerate pebbles are mainly dominated by hybrid sandstones and alveolina limestones. However, granitoids, diorites, basic volcanic rocks (Permian), siliciclastic red sandstone and microconglomerate (Buntsandstein facies), white quartz, green quartzite, and radiolarite are also common.

*4.2. Heavy Minerals*

Seventeen different transparent heavy minerals were successfully identified with the aid of Raman spectroscopy. Apatite (Ap), zircon (Zrn), tourmaline (Tur), rutile (Rt), epidote (Ep), titanite (Ttn), grossular (Grs), almandine (Alm), and staurolite (St) are the most abundant, whereas other transparent heavy minerals such as monazite (Mz), xenotime (Xtm), clinopyroxene (Cpx), spinel (Sp), sphalerite (Sph), chloritoid (Cld), andalusite (And), and kyanite (Ky) are scarce.

4.2.1. Heavy-Mineral Suites

We use correspondence analysis in order to explore similarities between the heavy-mineral content of samples and to define distinct heavy-mineral assemblages. Based on the results of the correspondence analysis (Figure 7), and the clear differences observed in the relative abundance of the heavy minerals (Figure S7), five different heavy-mineral suites can be labeled, based on the relative enrichment in Ap, ZTR (Zrn+Tur+Rt), Ep, St, Ttn, Grs, Alm, and other transparent heavy minerals (OtHM). The correspondence analysis (Figure 7a) provides evidence of the occurrence of an Ep+St+Ttn suite ("Ep dominated" suite), an Ap+ZTR suite ("Ap+ZTR dominated" suite), and a Grt+OtHM suite. However, this last mineral assemblage can be subdivided into a "Grs enriched" suite, an "Ep+St+Grs enriched" suite, and a "Ttn (+Grs +/−Ep) enriched" suite (Figure 7c), based on Grs, Ep, Ttn, St, and OtHM content.

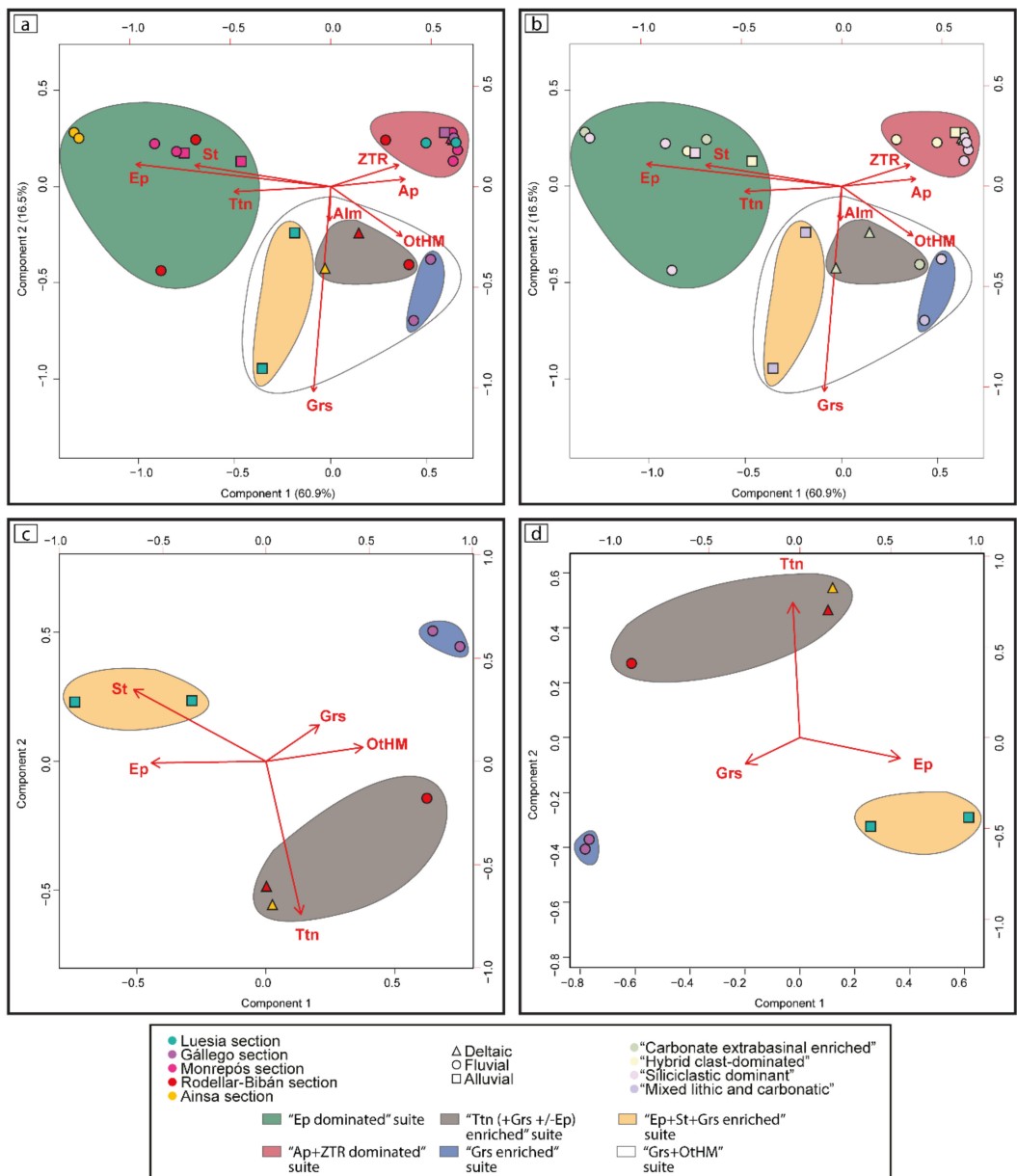

**Figure 7.** Heavy-mineral compositional plots. (**a**) Biplot displaying the results of correspondence analysis applied to all analyzed samples and considering all encountered minerals. OtHM include scarce minerals (Mz, Xtm, Cpx, Sp, Sph, Cld, And, and Ky). Colored symbols indicate sample's section. (**b**) Biplot displaying the results of correspondence analysis applied to all analyzed samples. Colored symbols indicate the petrofacies. (**c**) Biplot displaying the results of correspondence analysis applied to samples belonging to the Grs+OtHM enriched heavy-mineral suite only considering Grs, Ttn, Ep, St, and OtHM. (**d**) Biplot displaying the results of correspondence analysis applied to samples belonging to the Grs+OtHM enriched heavy-mineral suite only considering Grs, Ttn, and Ep.

"Grs (+Ttn +/−Ep) Enriched" Suite

This is the oldest mineral suite recorded in the study area, and it is dominated by ZTR (26.7–49.1%) and Ap (22.1–23.8%). However, it shows an important enrichment in Grs (12.1–16.0%), Ttn (5.6–13.8%), and Ep (0.0–12.0%) (Figure S7). The most characteristic feature of this assemblage is its Ttn content (Figure 7c,d). This suite occurs in the oldest deposits of the Ainsa and Jaca basins (the Sobrarbe delta, the Belsué-Atarés delta, and the lower fluvial Campodarbe Formation; Figures 7c and 8a) and is restricted to the easternmost sector of the study area.

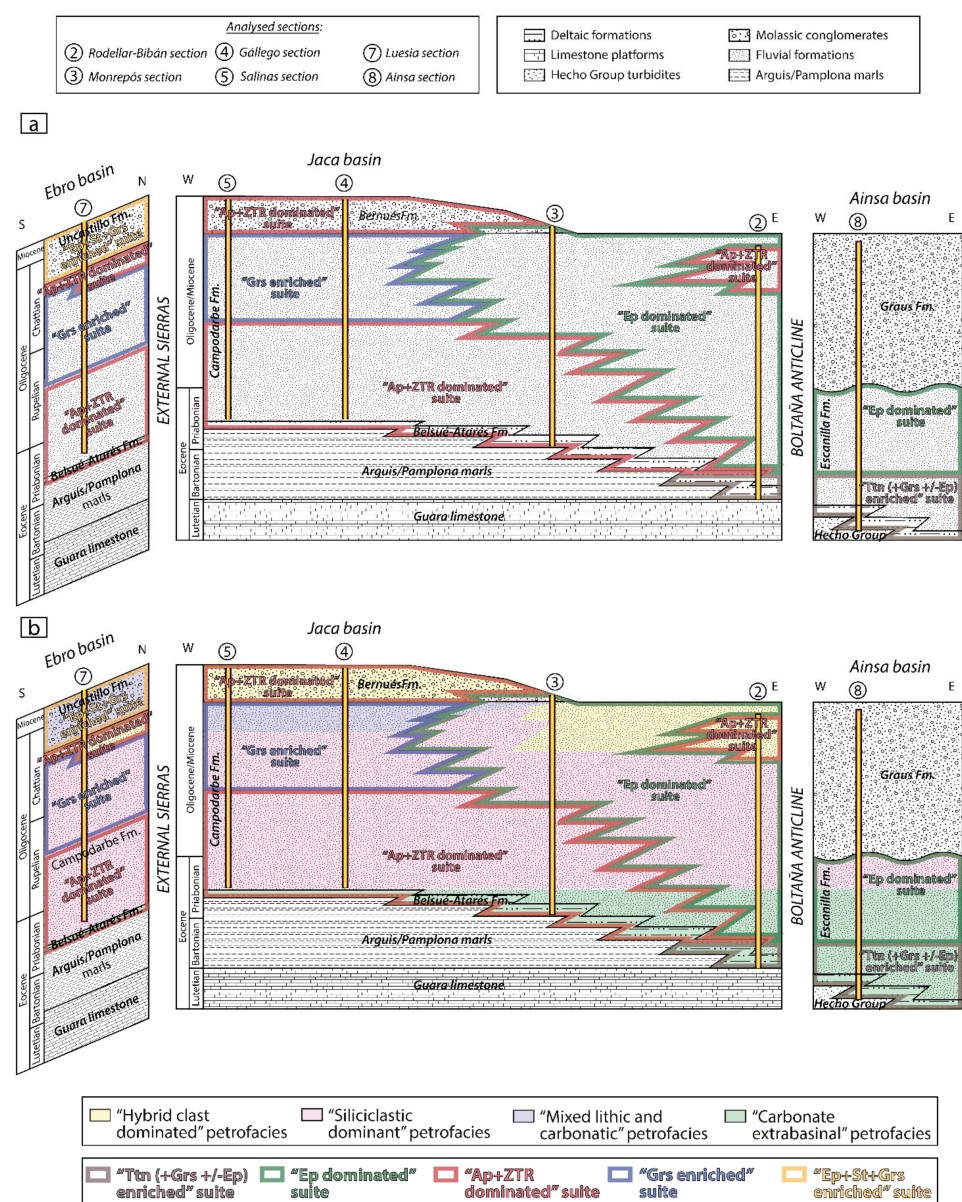

**Figure 8.** (**a**) Colored stratigraphic cross-section in order to illustrate the distribution of the heavy-mineral suites laterally and through time. (**b**) Colored stratigraphic cross-section in order to illustrate the distribution of the petrofacies and heavy-mineral suites laterally and through time.

### "Ap+ZTR Dominated" Suite

The distinctive feature of this suite is the dominance of Ap and ZTR. Together, they represent more than 87.7% of the mineral spectra, reaching up to 98.1% in the middle Campodarbe Formation in the Monrepós section (BEMO-8). Although OtHM (such as Cld, Mz, Xtm, or Sp) are scarce, this suite shows an enrichment in this component. The suite occurs in the Belsué-Atarés deltaic formation in the Nocito area (west of the Rodellar section) and dominates the lower and middle Campodarbe Formation in the Monrepós and Gállego sections, as well as the Bernués Formation in the Gállego area (Figure 8a). In the Ebro basin, it is present along the whole sedimentary record of the Campodarbe Formation. By contrast, in the Ainsa basin, this assemblage is not represented.

### "Ep Dominated" Suite

The main feature of this assemblage is the overwhelming presence of Ep, always higher than 40.5%, and reaching up to 77.4% in the lower Escanilla Fm in the Ainsa basin

(Figure 8a). This suite is found only in the eastern sector of the study area (Ainsa and eastern Jaca basin), dominating the fluvial Escanilla and Campodarbe Formations, but interestingly, it does not extend to the equivalent units of the western part of the Jaca basin (Gállego and Salinas sections). Its first appearance in the analyzed sections occurs in the upper Bartonian in the Ainsa and Bibán sections, whereas at Monrepós, it is not recorded until the Oligocene (upper Campodarbe).

"Grs Enriched" Suite

Although this suite is mainly dominated by Ap and ZTR (70.5–79.5%; Figures S7 and 7c,d), its remarkable feature is that it displays an enrichment in Grs (14.5–23.3%), Ep is absent, and Ttn scarce (0.5%). It occurs in the middle-upper Campodarbe deposits of the Gállego, Salinas, and Luesia sections (Figure 8a).

"Ep+St+Grs Enriched" Suite

This suite displays a Grs content (13.7–30.8%) similar to that of the Grs enriched suite (Figure S7), but it is always accompanied by remarkable proportions of Ep (24.0–27.6%; more than the Grs+Ttn+/−Ep enriched suite; Figure 7d) and St (Figure 7c); Ttn is scarce (1.5–1.9%). Ap+ZTR contents (33.9–53.9%) are much lower than in the Grs enriched and the Ap+ZTR dominated suites. The suite is only recorded in the Ebro basin, in the alluvial deposits of the Luesia fan (top of the Luesia section).

## 5. Discussion

### 5.1. Considerations about the Heavy-Mineral Detrital Signatures

The original detrital heavy-mineral suites derived from a source rock might be altered through a series of processes. Among them, dissolution during deep burial diagenesis is the most critical in old sedimentary rocks [130–132,145–147]. Therefore, it is important to assess the possible diagenetic overprint prior to extract potentially biased conclusions about the provenance and connectivity of the studied deposits. In the Jaca basin, the sharp appearance of epidote in the record, the low paleotemperatures experienced by the studied sedimentary rocks (<50 °C) [32,148], and the absence of advanced dissolution features in the more unstable minerals [149], such as epidote or titanite, point to a very low impact of diagenesis on the detrital heavy-mineral suites. Therefore, we can infer that heavy-mineral data are reflecting provenance features instead of burial-related overprint.

### 5.2. Provenance Implications and Evolution of Source Areas

Sandstone detrital modes and heavy minerals in the Jaca basin indicate more than a single source area, evidencing the interplay between fluvial–alluvial systems of diverse provenance from Eocene to Miocene times. The four petrofacies and the five heavy-mineral suites established in this study allow a better discrimination of these source areas, as well as the identification of different sediment routings and their evolution through time. These inferences are here discussed in terms of provenance and connectivity, based on an integrated approach that encompasses the petrofacies and the heavy-mineral suites.

The Belsué-Atarés Formation represents the first deltaic unit registered in the southern border of the Jaca basin (Figure 8). "Carbonate extrabasinal enriched" petrofacies is distinctive on this formation. The petrographic signatures are characterized by a dominance of carbonate grains, with subordinate plutonic rock fragments. Together with paleocurrents and facies architecture [23,59], these petrographic signatures support a provenance from the east, in the central Pyrenees. This source area would have included the Paleozoic basement of the Axial Zone, which provided significant amounts of plutonic components, together with Mesozoic and Paleocene limestones, which mainly delivered a wide range of carbonate grains. The compositional similarity ("carbonate extrabasinal enriched" petrofacies with "Ttn+Grs(+/−Ep) enriched" suite) between the Sobrarbe (Ainsa) and Belsué-Atarés (Jaca) Formations implies that they were connected during Lutetian–Bartonian times (Figure 8b).

During the Bartonian, this source continued supplying sediment to the Ainsa basin, feeding the fluvial Escanilla Formation, and transferring sediment to the Campodarbe and Belsué-Atarés Formations in the eastern part of the Jaca basin. From the middle Bartonian onwards, the Escanilla Formation displays the "Ep dominated" suite, which can be traced to the Jaca basin in its easternmost part (Figure 8). The main characteristic of this assemblage is the overwhelming content of epidote, pointing to a source with abundant Triassic dolerites [39,118]. Compositional similarity between the Campodarbe Formation and the Escanilla Formation (both formations display CEE petrofacies and Ep dominated suite) implies connectivity between the two fluvial units.

However, it is remarkable that, during the Bartonian, west of the Rodellar-Bibán section, the Campodarbe Formation displays the other distinct heavy-mineral suite, the "Ap+ZTR dominated". Idiomorphic Ap and ZTR in this suite can be linked to granitic sources [39], whereas the more rounded Ap and ZTR grains can be attributed to the recycling of the siliciclastic Mesozoic (i.e., Vallcarga Fm. [150]) and Paleocene sedimentary cover (i.e., Tremp Fm. [43]), but the lack of epidote is evidence of the lack of Triassic dolerites in the source area. The occurrence of the two different heavy-mineral suites in different outcrops of the Campodarbe Formation with the same petrofacies points to the contribution by two distinct fluvial systems following different routings, one feeding the eastern part of the basin, sourced from the central Pyrenees (where abundant Triassic dolerites occur), and the other feeding the western part, sourced from the eastern Pyrenees.

During the Priabonian, paleocurrent directions and facies architecture continue indicating an eastern provenance for the Campodarbe Formation, but a change of the petrofacies typology (from CEE to SD) is evidence of a shift in the source area, marked by an increase in metamorphic rock fragments. This shift indicates a persisting input from the Paleozoic basement but highlights a major change in the sourcing lithologies within the Axial Zone. Nonetheless, the new petrofacies (SD) still displays the two former heavy-mineral suites ("Ep dominated" and "Ap+ZTR dominated"; Figure 8b). The "Ep dominated" suite is evidence of the continued presence of Triassic dolerites in the source area, whereas the "Ap+ZTR dominated" suite points to the lack of these rocks. The dominance of Ap and ZTR in the heavy mineral provenance signal, together with the abundance of metamorphic rock fragments and siliciclastic sandstone, can be related to sources with a very low to low degree of metamorphism, as well as to the recycling of Carboniferous and Permo–Triassic siliciclastic sandstones [39].

The change from the CEE to the SD petrofacies is also identified in the upper part of the Escanilla Formation, in the Ainsa basin (Ainsa section), as well as in the Campodarbe Formation, in the Jaca basin (Rodellar-Bibán and Monrepós sections; Figure 6). Therefore, these sections record the evolution of the central and eastern Pyrenean sources, from a plutonic dominated toward a metamorphic dominated source area, that could be linked to a reorganization of the drainage area in the Axial Zone caused by uplift or thrust emplacement in the source area [30,151–153]. However, the compositional difference recorded by the heavy-mineral detrital signatures highlights the persistence of the two different axially-fed east-sourced systems from late Bartonian (Figure 8).

Moreover, during the Priabonian, the northeastern part of the basin (San Felices section) records the onset of north-sourced sediments, evidenced by the substitution of the "carbonate extrabasinal enriched" for the "hybrid clast-dominated" petrofacies. By contrast, the advent of north-sourced sediments to the southern margin of the Jaca basin (transverse-fed system) (Figure 6) takes place during the Oligocene as shown by the substitution of the "siliciclastic dominant" petrofacies (associated with the eastern provenance of the axially-fed systems) by the "hybrid clast-dominated" petrofacies (transverse system).

As in the former described petrofacies, the "hybrid clast-dominated" petrofacies displays two distinct heavy-mineral suites. The "Ap+ZTR dominated" suite can be attributed to the recycling of the Eocene turbidite basin [39] (north of the Jaca basin). By contrast, the "Ep dominated suite" is evidence of the strong eastern contribution and the mixing of the axial and transverse systems in the eastern part of the basin. This mixing is also

evidenced by the occurrence of the "mixed lithic and carbonatic" petrofacies at the top of the Monrepós section, resulting from the interplay of the "hybrid clast-dominated" petrofacies with the "siliciclastic dominant" petrofacies.

In the western part of the basin (Gállego and Salinas sections), the mixing between the axial and transverse systems is also registered by the "mixed lithic and carbonatic petrofacies, which displays a "Grs enriched" suite (Figure 8), evidencing the interplay between the "siliciclastic dominant Grs enriched" axial system and the "hybrid clast-dominated Ap+ZTR dominated" transverse system.

Therefore, in the Jaca basin, the different heavy mineral provenance signatures recorded in the eastern and western areas allow to characterize two axially-fed, east-sourced fluvial systems that coexisted during the sedimentation of the Campodarbe Fm, which is at variance with the classical sedimentological model of a unique fluvial system transferring sediments from east to west [23]. The fluvial Campodarbe Fm displays coarser grain-sizes in the eastern part of the basin and finer grain sizes to the west. This shift was interpreted as a facies change of the same fluvial course, with its proximal facies located to the east and the distal to the west. Nevertheless, the different heavy mineral provenance signatures reveal that, in fact, this facies change corresponds to two different fluvial networks with different source areas instead of to proximal-distal parts of a unique fluvial system. The easternmost fluvial system ("Ep dominated") could correspond to the Bibán fluvial facies of Puigdefàbregas [23], which laterally passes northwest to the fluvio–lacustrine facies of Santa Cruz-Bailo. Meanwhile, the westernmost fluvial system ("Ap+ZTR dominated") would correspond to the Monrepós-Anzánigo fluvial facies, laterally passing to the west to the fluvio–lacustrine facies of Javier-Pintano-Villalangua.

Much more to the west, in the Luesia section, the lower Campodarbe Formation does not show significant variations with its equivalents located in the Jaca basin. At the upper part, it records the irruption of the transverse-fed system characterized by the "hybrid clast dominated" petrofacies and the "Ap+ZTR dominated" suite (Figure 8b). To the top, an abrupt provenance change is recorded by the Oligo–Miocene alluvial deposits of the Luna fan (Uncastillo Formation, in the Ebro basin), evidenced by the occurrence of the "mixed lithic and carbonatic" petrofacies and the "Ep+St+Grs enriched" suite, only recorded in these alluvial deposits (Figure 8b). This provenance signature contrasts with its time equivalent Bernués Formation, which characterizes the north-derived alluvial sedimentation more to the east (Gállego and Salinas sections). Although some components are common, the relatively high content of siliciclastic sandstone, radiolarite, and volcanic rock fragments allow to infer a distinctive source area for the conglomerates of the Luna fan, which is also highlighted by the enrichment of epidote, titanite, and grossular in the heavy-mineral detrital signatures.

It follows that the abundant siliciclastic content of the Luna fan cannot be derived from the same source areas of the hybrid sandstone and carbonate-rich San Juan de la Peña and Peña Oroel fans (Bernués formation), located straight north of the Luesia area [30]. Hence, we propose that the source area of the Luna fan was located in the western Pyrenees, in the Paleozoic Basque massifs (Figure 1), which account for this distinctive petrologic signature. The work by Hirst and Nichols [53] also pointed to this western source, based on heavy mineral data from the Luna fan. All these are in agreement with thermochronological data from the Axial Zone of the western Pyrenees, which show older exhumation ages than the Axial Zone of the west-central Pyrenees [154–156].

The interpretation of the "mixed lithic and carbonate" petrofacies recorded in the Martés sandstone in the northwestern part of the Jaca basin is more challenging. The siliciclastic content of this unit is very close to the composition described in the Luna alluvial fan, which could indicate the same source area for both, located to the northwest, in the western Pyrenees. This interpretation would discard an eastern source area from the central Pyrenees, as assumed before, according to the low content on metamorphic rock fragments. Nonetheless, this interpretation is not in accordance with north-west paleocurrent directions reported for the Martés sandstone by Puigdefàbregas [23], which imply a north-west

directed paleoflow, probably derived from eastern source areas according to late Eocene basin paleogeography.

### 5.3. Functioning of the Sediment Routing Systems

During late Lutetian to Bartonian times (Figure 9a), deltaic sedimentation in the southern Jaca basin was mainly derived from eastern source areas. These sources were the Paleozoic basement and the Mesozoic and Paleogene sedimentary cover of the growing central Pyrenees. The Paleozoic source rocks contributed by delivering plutonic components trough a unique fluvial system for the first stage of deltaic sedimentation (late Lutetian to Bartonian). However, from the late Bartonian onwards, heavy-mineral detrital signatures reveal that two distinct axially-fed fluvial systems from the central-east Pyrenees were delivering sediment to the basin; one dominated by epidote and the other dominated by ultrastable apatite, zircon, tourmaline, and rutile (Ap+ZTR). The first one fed the easternmost sector of the Jaca basin through the Escanilla sediment routing system [28]. The other extended its influence to the westernmost area and must have entered the basin following a more meridional sediment routing through the Priabonian Salinar and/or the Rupelian lower Peraltilla Formations of the autochthonous foreland basin [157], which are characterized by absent or scarce epidote (Figure 9b). The axially-fed system enriched in epidote is sourced from the central Pyrenees, from the same source areas as the Sis and Gurp alluvial fan conglomerates, where Triassic dolerite rock fragments (bearing abundant epidote) are frequent (northern part of the Central South Pyrenean Unit), with abundant Keuper diapiric occurrences [158–160]. However, the Ap+ZTR dominant system must be sourced from a farther, more eastern sector of the southern Pyrenees, where Triassic dolerites of the Keuper facies are less abundant or absent (Pedraforca-Port del Comte area and present-day Segre Valley, Eastern Pyrenees).

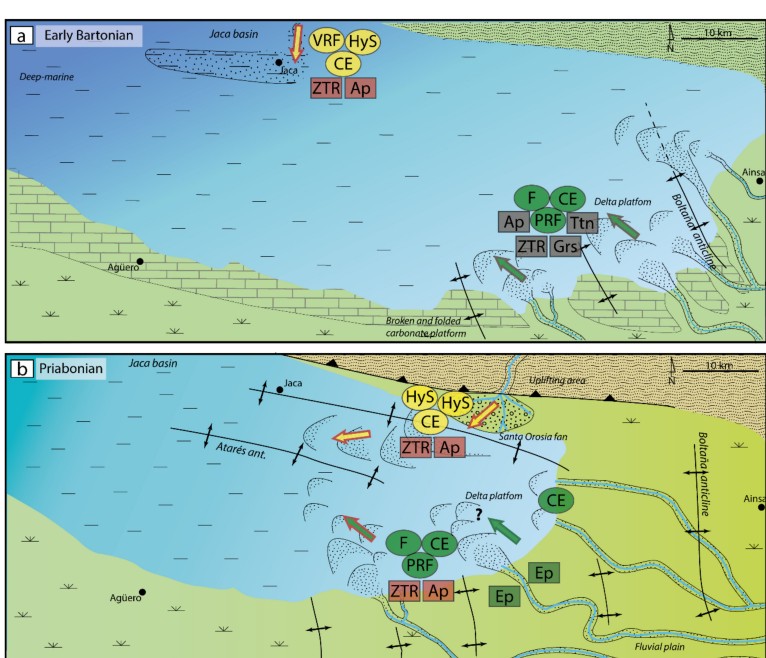

**Figure 9.** Paleogeographic scheme of the Jaca basin during Bartonian–Priabonian times. (**a**) Early Bartonian. (**b**) Priabonian. Circles and squares highlight distinct components. PRF: plutonic rock fragments, F: feldspar grains, CE: carbonate extrabasinal grains, HyS: hybrid sandstone rock fragments, VRF: volcanic rock fragments, Ap: apatite, ZTR: ZTR, Ep: epidote, Grs: grossular, Ttn: titanite. Circle colors correspond to petrofacies described in Figure 4. Square colors correspond to heavy-mineral suites described in Figure 7. Arrows indicate petrofacies (fill) and heavy-mineral suite (stroke). Reconstruction of the maps based on Puigdefàbregas, Bentham et al., Hogan, Montes, Caja et al., Huyghe et al., Roigé et al., Boya, Coll et al. [23,24,27,29,30,39,63,67,72,76,97].

Both differentiated fluvial systems persisted but evolved to a more dominant metamorphic composition that persisted from middle Priabonian until at least Chattian–Aquitanian times (Figure 10a), as demonstrated by the compositional features of the upper Campodarbe Formation.

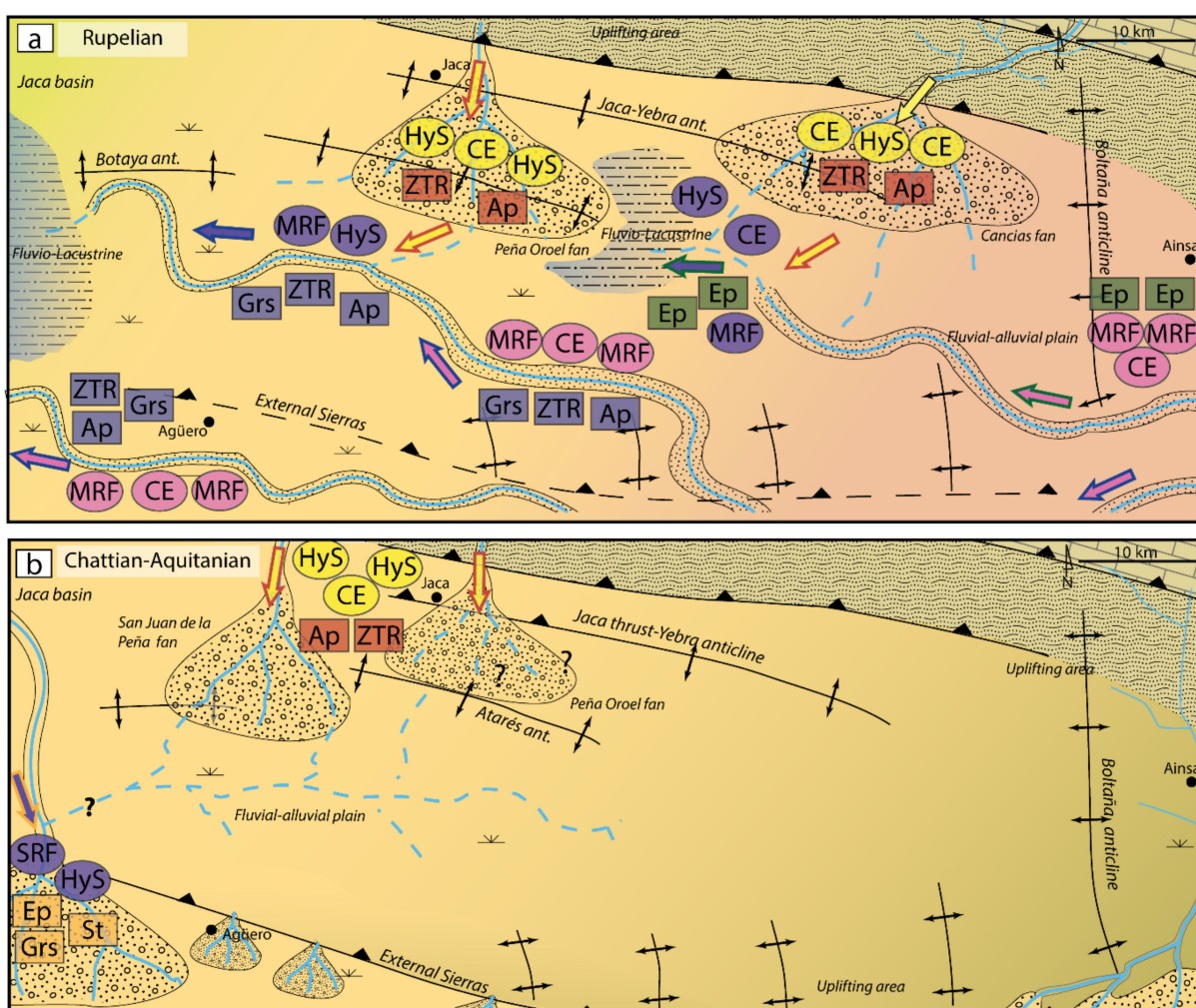

**Figure 10.** Paleogeographic scheme of the Jaca and Ebro basins during Rupelian–Aquitanian times. (**a**) Rupelian. (**b**) Chattian–Aquitanian. Circles and squares highlight distinct components. MRF: metamorphic rock fragments, SRF: sedimentary rock fragments, CE: carbonate extrabasinal grains, HyS: hybrid sandstone rock fragments, Ap: apatite, ZTR: ZTR, Ep: epidote, Grs: grossular, Ttn: titanite, St: staurolite. Circle colors correspond to petrofacies described in Figure 4. Square colors correspond to heavy-mineral suites described in Figure 7. Arrows indicate petrofacies (fill) and heavy-mineral suite (stroke). Reconstruction of the maps based on Hirst and Nichols, Puigdefàbregas, Friend et al., Arenas, Nichols and Hirst, Jones, Roigé et al., Boya, Coll et al. [23,29,30,39,53,76,97,99,161–163].

Moreover, during this time, northern sources, mainly composed by the Eocene Group turbidites, were uplifted in the northern Jaca basin by the activity of the Gavarnie thrust. These source areas led to progressive mixing and finally caused the replacement of the two distinct axial-fed fluvial systems by transverse north-derived alluvial systems, producing a westward and southward displacement of the axial fluvial network.

The later stages of the Jaca basin (Figure 10b) were determined by the activity of the Guarga thrust that produced the uplift of the basin margins (External Sierras) preventing the axial fluvial network to enter the basin. The Chattian to Aquitanian period was mainly characterized by two north-derived systems from different source areas. For the Bernués

Formation (Jaca basin), the source area was situated immediately to the north of the basin, composed by the Hecho Group turbidites and the North Pyrenean Zone [30], although in the eastern sector the influence of the east-derived systems persisted in the first stages of the sedimentation. In contrast, the source area for the Uncastillo Formation (Ebro basin) was located in the western Pyrenees, composed by the Paleozoic basement Basque massifs and the earlier foreland deposits (Hecho Group and Campodarbe Formations).

## 6. Conclusions

The combination of new sandstone petrography and heavy-mineral data allowed to constrain the interplay of diverse sediment routing systems in the transitional to terrestrial environments of the Jaca thrust-sheet-top basin and the Ebro autochthonous basin during Lutetian to Miocene times. Additional data of the equivalent sedimentary systems of the Ainsa basin more to the east allowed for a better characterization of the evolution of eastern source areas.

Deltaic sedimentation in the southern Jaca basin (Bartonian–Priabonian) was mainly derived from eastern source areas, located in the central Pyrenees, in which the Paleozoic basement contributed by delivering dominant plutonic components.

During Priabonian times, the Campodarbe Formation records a change in the source area that yielded to the evolution of a Paleozoic source richer in metamorphic rocks, a signature that persisted until at least Oligo–Miocene times. However, heavy-mineral detrital signatures evidence that two distinct major fluvial systems coexisted, one sourced from the central Pyrenees and the other from the eastern Pyrenees, delivering sediment to different parts of the basin. This differentiation (Figure 11) could not have been made without the aid of the heavy mineral analysis.

On the other hand, north-derived transverse fluvial systems eventually replaced the axial systems, progressing from east to west. In the Chattian to Aquitanian record, two main north-derived systems can be distinguished in the Jaca and Ebro basins. Whereas in the Jaca basin, the Bernués Formation came from source areas comprising the North Pyrenean Zone and, more importantly, the uplifted Eocene foreland basin, the coetaneous Uncastillo Formation of the Ebro basin was sourced from the western Pyrenees, comprising the Paleozoic Basque massifs and also by previous foreland deposits.

This work highlights how the integration of sandstone petrography and heavy mineral analysis provides a higher resolution to characterize the evolution of sediment routing systems from a "source-to-sink" approach. Our study provides evidence that the coupling of these techniques is a much more powerful tool that can resolve aspects of the routing systems that could not be disentangled by one method alone. Although, nowadays, the integration of sandstone petrography and heavy mineral analysis is considered as time-consuming and rarely used, this work argues that it is the best approach to fully characterize source-to-sink relationships in a clastic sedimentary basin.

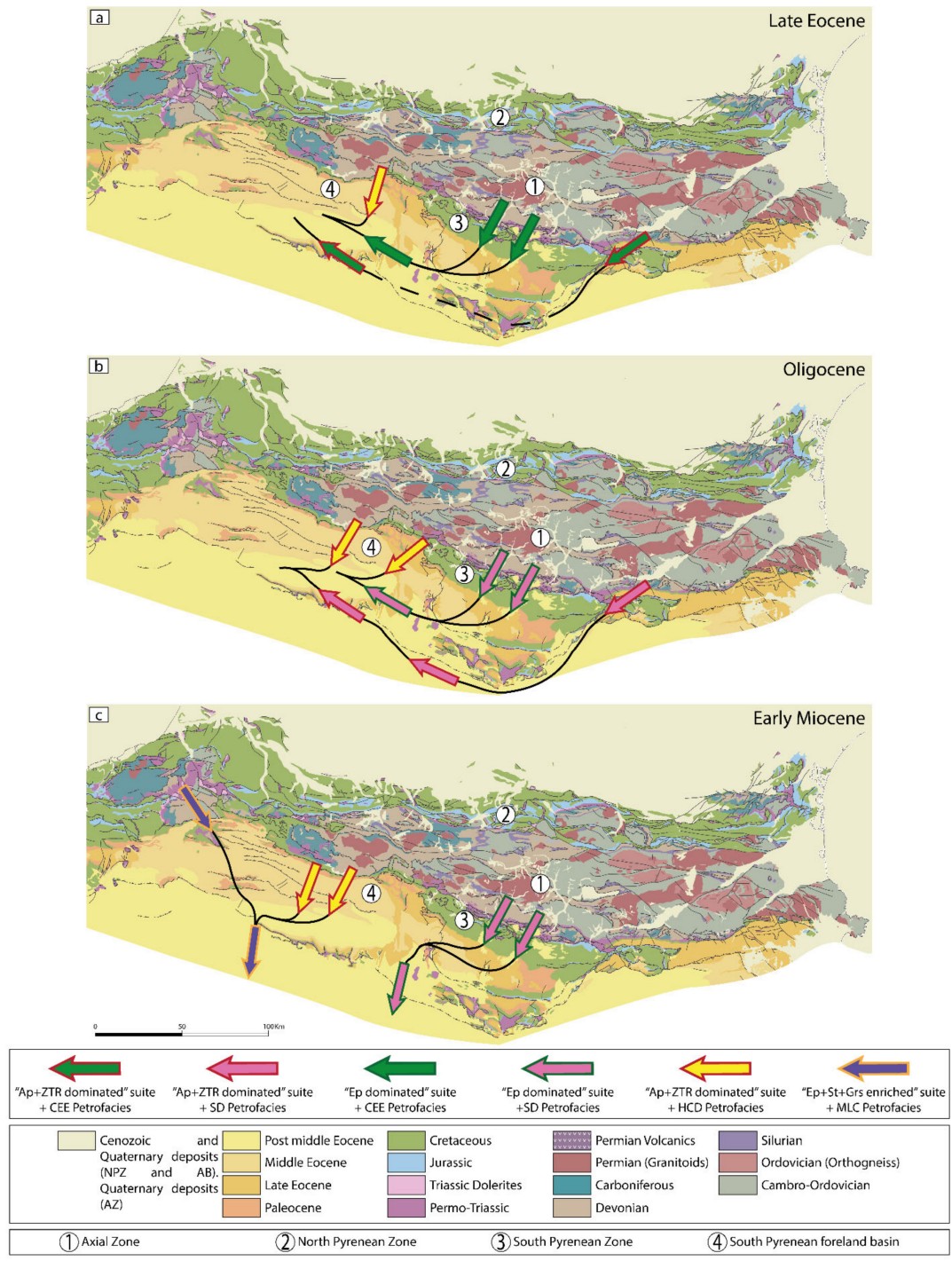

**Figure 11.** Summary of source areas and sediment routings in the South Pyrenean Basin from late Eocene to early Miocene displayed in a present-day map, not restored. Coarse arrows indicate composition of source areas (stroke color: heavy-mineral suite, fill: petrofacies). Thin arrows indicate sediment routing approximately. (**a**) Late Eocene routing systems. Two different axially-fed east sourced systems (one more meridional) coexist with a transverse-fed north sourced system supplying the Jaca basin. (**b**) Oligocene routing systems. Two different axially-fed east sourced systems (one more meridional) coexist with two transverse-fed north sourced systems supplying the Jaca basin. (**c**) Early Miocene routing systems. Two different transverse-fed north sourced systems supply sediment to the Ebro basin in the western area, whereas in the eastern part, two transverse-fed north sourced systems supply the Huesca fan.

**Supplementary Materials:** The following information is available online at https://www.mdpi.com/article/10.3390/min12020262/s1, Figure S1: Representative Raman spectra of heavy minerals from the late Eocene–Miocene Jaca basin, Figure S2: Optical photomicrographs of distinct extrabasinal grains, Figure S3: Compositional plots for all the analyzed samples, Figure S4: Images of the appearance of each of the described petrofacies in the conglomerate-sized deposits, Figure S5: Biplot of the two sub-groups of the MLC petrofacies, Figure S6: Heavy-mineral percentage pie charts for all the analyzed samples in a general stratigraphic cross-section sketch, Figure S7. Heavy-mineral percentage pie charts for all the analyzed samples in a general stratigraphic cross-section sketch. Table S1: Sandstone compositional data (point counting analyses), Table S2: Heavy-mineral compositional data (point counting analyses), Sample coordinates: Geographic coordinates of the studied samples.

**Author Contributions:** Conceptualization, D.G.-G., M.R., X.C., S.B., and A.T.; methodology, D.G.-G., M.R., X.C., and N.M.; formal analysis, X.C., and M.R.; investigation, X.C., M.R., D.G.-G., and S.B.; data curation, X.C., M.R., D.G.-G., and S.B.; writing—original draft preparation, X.C., and M.R.; writing—review and editing, X.C., M.R., D.G.-G., A.T., S.B., and N.M.; visualization, X.C, M.R., and D.G.-G.; supervision, D.G.-G., M.R., and A.T.; project administration and funding acquisition, A.T. All authors have read and agreed to the published version of the manuscript.

**Funding:** This research was funded by the projects CGL2014-54180-P and PGC2018-093903-B-C21, financed by the Ministerio de Economia y Competitividad (MINECO) and Ministerio de Ciencia, Inovación y Universidades (MCIU) of Spain. X. Coll acknowledges support from the Ministerio de Cultura, Deporte y Educacción (MECD) of Spain (FPU16/00219).

**Institutional Review Board Statement:** Not applicable.

**Informed Consent Statement:** Not applicable.

**Data Availability Statement:** The data presented in this study are available in the Supplementary Materials here.

**Acknowledgments:** We are very grateful to the anonymous reviewers for providing constructive reviews that helped to improve the original paper.

**Conflicts of Interest:** The authors declare no conflict of interest. The funders had no role in the design of the study; in the collection, analyses, or interpretation of data; in the writing of the manuscript; or in the decision to publish the results.

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
