# Peer review of "Interplay of Multiple Sediment Routing Systems Revealed by Combined Sandstone Petrography and Heavy Mineral Analysis (HMA) in the South Pyrenean Foreland Basin"

_minerals, doi:10.3390/min12020262_

Round 1
Reviewer 1 Report
Dear Authors,
Please see the attached PDF file for comments.
Regards

Reviewer 2 Report
The paper is a member of the common studies concerned with provenance analysis, encompassing HM and lithoclasts in context with a review of the HM-hosting environments of deposition. This is a good approach. The application of Raman is a modern but not really new one and as such I would like to have seen a bit more on the determination of the HM and also raise the question why LM and opaque minerals have been excluded (see, e.g.,MAFTEI A.E., BUZATU A., DAMIAN G., BUZGAR N., DILL H.G. and APOPEI A.I. (2020) Micro-Raman - a tool for the heavy mineral analysis of gold placer-type deposits (Pianu Valley, Romania).- Minerals 10: 10.3390/min10110988
Author Response
Authors: Although we couldn’t find any guides on how to improve the manuscript according to the general comments by reviewer 2, we have tried to improve it incorporating the following changes:
- R2: English language and style: Moderate English changes required
- Authors: We modified the manuscript according to reviewer 1 English minor spell checks suggestions. However, we have revised our English accurately again and, as reviewer 1 states we agree that is fine.
- R2: Does the introduction provide sufficient background and include all relevant references? Must be improved.
- Authors: As reviewer 2 suggests, we have thoroughly read again the introduction. Anyway, reviewer 2 does not point out which part of the introduction needs improving or what points are missing. As reviewer 1 suggests, we believe it provides sufficient background about provenance studies, heavy-minerals, the Pyrenees, the South Pyrenean basin, and the Jaca basin. Hence, we agree with Reviewer 1.
- R2: Is the research design appropriate? Can be improved.
- Authors: We understand that this refers to R2 comment “The application of Raman is a modern but not really new one and as such I would like to have seen a bit more on the determination of the HM and also raise the question why LM and opaque minerals have been excluded”.
Further characterization of LM, HM, and opaque minerals with Raman spectroscopy is beyond the scope of this paper. We thoroughly read the paper recommended by reviewer 2 and find it very interesting on how to characterize the geochemistry of heavy minerals. We are familiarized with this kind of studies, and indeed, we have differentiated between Almandine and Grossular garnet by using Raman spectroscopy. However, further characterization of the geochemistry of other mineral phases is beyond the scope of this basin-scale paper.
Our approach is to study LM by optical microscope through point-counting standard procedures widely-used in sedimentary provenance analysis. Furthermore, we used Raman spectroscopy for heavy mineral identification, avoiding subjectivity from microscope operator. As for opaque minerals, we know Raman can be used to identify opaque minerals, but it also has its limitations. In fact, we used this approach in a previous work (Coll et al. 2016), but opaque minerals did not bring any relevant conclusions in terms of sedimentary provenance analysis in the South Pyrenean basin, which is the aim of our work.
However, we think the data presented in this paper is original and relevant for the scientific community, since this kind of studies are scarce in the South Pyrenean basin. Our data provide petrographic signatures of the study area that area have never been characterized allowing a better constraining of the South Pyrenean basin sediment-routing systems.
- R2: Are the methods adequately described? Must be improved.
- Authors: We believe the methods used here (Point-counting, heavy-mineral separation, statistical analysis) have been thoroughly described, and clearly referenced for the readers who need more information about the methods. We agree with reviewer 1 since we described the methods as usual as in our previous works (Roigé et al., 2016; Roigé et al., 2017; Gómez-Gras et al., 2017; Roigé et al., 2019; Coll et al., 2020).
- R2: Are the results clearly presented? Must be improved
- Authors: The results of the statistical treatment used to define our petrofacies and heavy-mineral assemblages (the relevant information), which is the purpose of this work, are clearly presented in high-quality figures trough the manuscript. Nowadays, correspondance analysis (CA) and biplot figures are widely-used to display this kind of results in high impact geology journals, and taught and recommended in international courses dealing with this approaches such as the Heavy Mineral School (University of Milano Bicocca) or Compositional Data Analysis (Universitat de Girona). In addition, the results are incorporated into regional stratigraphic sketches and paleogeographic reconstructions, all these helping to understand de compositional characterization of the analyzed deposits and the implications of our results. In addition, in the supplementary material we added complementary information (clearly presented and explained) that include useful data and figures for researchers in the field of “Source to Sink” and sedimentary provenance analysis of the South Pyrenean basin.
- R2: Are the conclusions supported by the results? Must be improved
- Authors: Our conclusions have been modified and improved, although they are well supported by the results. We agree with reviewer 1 that “At the moment there is too much detail that is completely unnecessary” and changes have been made accordingly.
Round 2
Reviewer 2 Report
I accept the comments by the authors